# Real-Time Aerial Multispectral Object Detection with Dynamic Modality-Balanced Pixel-Level Fusion

**DOI:** 10.3390/s25103039

**Published:** 2025-05-12

**Authors:** Zhe Wang, Qingling Zhang

**Affiliations:** 1School of Aeronautics and Astronautics, Shenzhen Campus of Sun Yat-sen University, No. 66, Gongchang Road, Guangming District, Shenzhen 518107, China; wangzh363@mail2.sysu.edu.cn; 2Shenzhen Key Laboratory of Intelligent Microsatellite Constellation, Shenzhen Campus of Sun Yat-sen University, No. 66, Gongchang Road, Guangming District, Shenzhen 518107, China

**Keywords:** aerial object detection, real time, full time, modality imbalance, multispectral fusion, Multispectral Luminance Weighted Fusion, Dynamic Modality Dropout and Threshold Masking

## Abstract

Aerial object detection plays a critical role in numerous fields, utilizing the flexibility of airborne platforms to achieve real-time tasks. Combining visible and infrared sensors can overcome limitations under low-light conditions, enabling full-time tasks. While feature-level fusion methods exhibit comparable performances in visible–infrared multispectral object detection, they suffer from heavy model size, inadequate inference speed and visible light preferences caused by inherent modality imbalance, limiting their applications in airborne platform deployment. To address these challenges, this paper proposes a YOLO-based real-time multispectral fusion framework combining pixel-level fusion with dynamic modality-balanced augmentation called Full-time Multispectral Pixel-wise Fusion Network (FMPFNet). Firstly, we introduce the Multispectral Luminance Weighted Fusion (MLWF) module consisting of attention-based modality reconstruction and feature fusion. By leveraging YUV color space transformation, this module efficiently fuses RGB and IR modalities while minimizing computational overhead. We also propose the Dynamic Modality Dropout and Threshold Masking (DMDTM) strategy, which balances modality attention and improves detection performance in low-light scenarios. Additionally, we refine our model to enhance the detection of small rotated objects, a requirement commonly encountered in aerial detection applications. Experimental results on the DroneVehicle dataset demonstrate that our FMPFNet achieves 76.80% mAP50 and 132 FPS, outperforming state-of-the-art feature-level fusion methods in both accuracy and inference speed.

## 1. Introduction

Aerial object detection, as a pivotal technology bridging machine vision and remote sensing, plays a critical role in military reconnaissance [1,2], disaster forecasting [3], resources exploration [4] and environmental monitoring [5]. Airborne platforms like drones and satellites have advantages related to their quick response across various situations due to their high flexibility, prompting the development of full-time and real-time detection models to be deployed on them. With the rapid and constant development of Convolutional Neural Networks (CNNs) and Transformers, deep learning-based methods have overwhelmed traditional machine learning methods [6,7,8] and exhibited satisfactory detection performance in various applications. Contemporary deep learning-based approaches predominantly rely on visible-light sensors that capture rich spectral and textural information under normal illumination [9,10,11]. However, these RGB-based methods exhibit severe performance degradation in low-light scenarios due to their inherent sensitivity to illumination conditions [12], particularly lacking reliable feature extraction at night or in adverse weather [13], leading to compromised performance in full-time object detection. To deal with this problem, infrared (IR) sensors were introduced to provide a fundamental solution. Unlike visible sensors relying on reflected light, IR sensors detect thermal radiation emitted by objects, enabling illumination-invariant detection capability [14]. The complementary characteristics of both modalities have inspired the development of RGB-IR multimodal detection systems—visible modalities excel in detailed feature representation under adequate light conditions, while infrared modalities ensure robustness in dark scenes. Equipping drones and satellites with dual-mode sensors enables the synergistic utilization of both detailed color and textural information and illumination-invariant thermal signals, thereby enhancing detection reliability across diverse situations [15,16]. While applying visible sensors during the day and applying infrared detection at night may achieve full-time object detection with decent accuracy, numerous studies have demonstrated that multispectral models significantly outperform unimodal RGB or infrared detection models by integrating cross-modality advantages [17,18,19]. However, this performance enhancement comes with addition computational cost, with the incorporation of IR data essentially increasing the input channels, posing more challenges for real-time processing deployed on resource-constrained edge devices like drones and satellites.

Currently, feature-level fusion methods are attracting the interest of most researchers and becoming mainstream since they can adequately make use of the complementarity of RGB and infrared characteristics. Feature-level fusion methods utilize a dual backbone network to primarily extract features of visible images and infrared images, respectively, then a custom-designed modality-fusion module integrates and combines the features extracted by backbone networks. The fusion outputs are multi-scale feature maps to be subsequently processed in model neck and head for detection. The overall structure of feature-level fusion is shown in Figure 1a. Driven by accuracy-centric architectures [20,21,22] with customized designs, feature level-fusion methods have spawned diverse variants, achieving comparable detection performances in aerial object detection. Nevertheless, existing multimodal frameworks face two critical limitations that hinder their practical airborne deployment:

Firstly, dual feature extraction backbones for separate modality processing and multi-scale multiple feature fusion modules increase the model complexity, leading to structural redundancy and computation overload, which conflict with real-time object detection in resource-limited airborne platforms. Features in different modalities will be extracted in detail in a separate branch, which is not always necessary, e.g., visible features in dark scenes, further resulting in false learning of invalid features.

Secondly, existing multispectral fusion methods mainly concentrate on complementary feature exploitation and unique modality feature enhancement, overlooking the impact of modality imbalances. It is expected that in daylight scenes, visible light provides more information and infrared features serve as auxiliary, while in low-light scenes, infrared features should take the lead. However, due to the visible modality’s richer spectral and textual features and the large amount of daylight scene training, visible features dominate while infrared images suffer from insufficient learning, and the visible modality can even interfere with infrared processing in dark environments [23]. As a result, in some extreme illumination scenes, multispectral detection models underperform against infrared-only models.

Considering the paradox between full-time and real-time in aerial object detection, our purpose is to minimize the model size increase and computation overhead when designing multispectral object detection methods, applying a model that achieves the best accuracy–speed trade-off, to realize full-time and real-time object detection in airborne platforms. To this end, we propose the Full-time Multispectral Pixel-level Fusion Network (FMPFNet). Firstly, inspired by calculating parameters of each stage of backbone, we propose a pixel-level fusion method named Multispectral Luminance Weighted Fusion (MLWF) with necessary accuracy-driven attention-based designs. Although complementary selection and modality attention and enhancement designs are effective, they introduce additional computation cost. Therefore, we apply them before the first CBS (convolution, batch normalization and Silu) feature extraction layer to minimize inference speed sacrifice. Figure 1b illustrates a pixel-level fusion method that simultaneously considers detection performance and model complexity. Furthermore, we propose Dynamic Modality Dropout and Threshold Masking (DMDTM), a simple yet effective pixel-wise augmentation strategy. This strategy improves full-time detection accuracy by moderating the imbalance between two modalities and enhances information exploitation of infrared images without introducing additional modules or harming inference speed. In addition, we choose YOLOv10s as our base model due to its optimal accuracy–speed trade-off among the YOLO series. To address the absence of an OBB head in the original YOLOv10 model, we design a dual label assignment OBB head inspired by YOLOv8 with a similar model structure to enable arbitrary direction object detection. We also improve the loss function for better small object detection. The results on the Drone-Vehicle dataset demonstrate that our model achieves state-of-art detection accuracy and advantaged inference speed compared with feature-level fusion-based models.

Contributions of our work can be summarized as follows:
We revisit pixel-level fusion by proposing the MLWF module, which contains a fine-designed two-tier cascade reconstruction sub-module, accomplishing real-time and full-time aerial object detection with satisfactory performances.We propose the DMDTM training augmentation strategy, which enables the model to balance attraction between both modalities and focus on infrared images in low-light conditions. To the best of our knowledge, we are the first to propose augmentation methods specialized for inherent characteristic imbalance between different modalities in multispectral object detection.We design an OBB head for the base YOLOv10s model and adjust the loss function for aerial small object detection. Our model exhibits an mAP of 76.80 and an FPS of 132, accomplishing competitive accuracy with outstanding speed among state-of-the-art models.

## 2. Related Work

### 2.1. Deep Learning-Based Object Detection Models

Deep learning-based object detection algorithms have become irreplaceable in the mainstream in computer vision since the proposal of ResNet [24]. Convolutional Neural Networks (CNNs) have evolved rapidly and played an essential role in recent years, with detection frameworks mainly divided into two categories: one-stage algorithms and two-stage algorithms. Two-stage algorithms consist of two independent network stages. The first stage generates candidate region proposals and the second stage performs classification and bounding box regression in these regions. The R-CNN series, including R-CNN [25], Fast-RCNN [26], Faster-RCNN [27], Mask-RCNN [28], etc., are typical representatives. One-stage methods, like SSD [29], Retinanet [30] and the YOLO [31] series, unify region proposal and classification into a single neural network, which are end-to-end architectures directly predicting bounding box coordinates and object classes. While two-stage algorithms achieve marginally higher accuracy, one-stage algorithms ensure their accuracy with lower computation cost, lighter model size and faster inference speed, making them more suitable for real-time aerial object detection.

With further research on one-stage algorithms, the YOLO series [31] has gained widespread attention for its real-time speed and high accuracy, achieving great advantages in tasks with limited computation resources. YOLO models consist of three parts: a backbone that extracts features, a neck (mainly composed of FPN and PAN networks) that fuses multi-scale features and a head that predicts bounding boxes and object classes. Since YOLOv4 [32], the architecture of the YOLO series has incorporated a series of optimizations across all components (backbone, neck, head), collectively enhancing not only detection performances but also deployment flexibility. They [33,34,35] support multiple size variants, providing lightweight versions of catering edge devices with limited computational resources. Compared with previous YOLO models, they enjoy flexible model size and higher inference accuracy due to constantly improving CSP [36] modules, which effectively strengthen feature extraction.

### 2.2. Multispectral Aerial Object Detection

The fusion methods are primarily grouped into three strategies based on the different stages, i.e., pixel-level fusion, feature-level fusion and decision-level fusion [37]. Decision-level fusion fuses the detection results at the final stage, giving results based on the respective modality, often suffering from high computation overhead caused by repeated calculations. In contrast with decision-level fusion, pixel-level fusion and feature-fusion have gained broader adoption.

Liu et al. [38] proposed vanilla ConvNet using Faster-RCNN as base model, indicating that halfway fusion, which performs fusion on the middle level, exhibits the best performance. This conclusion highlighted feature-level fusion, gathering increasing research interest in multispectral object detection. Early approaches focused on simple concatenation of features from different modalities that failed to adequately exploit the complementary information between modalities. To better extract features from the respective modalities and fuse them for subsequent object detection tasks, researchers have increasingly investigated attention mechanisms for sophisticated feature integration. Fang et al. [39] proposed an effective self-attention mechanism-based cross-modal feature fusion method to better utilize information in both modalities. Bao et al. [40] proposed the Attention Fusion Module and the Fusion Shuffle Module to efficiently process and integrate features from infrared and visible images, meanwhile introducing Fusion Loss to reduce required training epochs. Tang et al. [41] proposed the ConTriNet, which comprises a triple-flow architecture including two modality-specific flows and a modality-complementary flow with robustness against noise originating from defective modalities, meanwhile applying multiple modules to reduce modality imbalance and enhance multi-scale and multispectral feature capture and fusion through Divide-and-Conquer, yielding a high-quality saliency map for optimal RGB-Thermal Salient Object Detection. Li et al. [42] proposed a novel Transformer-based RGB-T MOT model utilizing a modality template to globally bridge the cross-modal interaction and adaptively fuse RGB and TIR search region tokens, enabling sufficient context modeling; then, TPTU strategy is applied to adequately leverage temporal contexts for subsequent tracking tasks via dynamic template updating. Aside from attention mechanisms, adaptive feature fusion methods have also come into view. GFD-SSD [43] combined feature maps from different modalities effectively by introducing Gated Fusion Units, which control the weight of features, performing dynamic adjustment based on training datasets. Zhang et al. [44] proposed Guided Attentive Feature Fusion, which combines intra-modality and inter-modality attention modules, empowering network fusion of multispectral features by dynamical weighting.

However, feature-level fusion models often contain two identical multi-scale feature extraction backbones to generate feature maps of the respective modalities with similar feature information, then apply feature fusion on multiple scales of feature maps, which doubles the computation costs of the feature fusion stage. Although they achieve acceptable performance, structural redundancy and heavy overall computation overload arise regardless of whether fusion module designs are lightweighted, which is not conducive to real-time aerial object detection.

Apart from feature-level fusion, pixel-level fusion methods have also drawn increasing attention from researchers. Pixel-level fusion methods exploit the complementary information of visible and infrared modalities and synthesize them into a four-channel image at the primary stage of feature extraction. Pixel-level fusion is widely applied in various image fusion scenes, e.g., multi-focus fusion [45], multi-exposure fusion [46], medical image fusion [47], visible–infrared fusion [48] and remote sensing pan-sharpening [49]. The majority of traditional pixel-level fusion approaches can be categorized given the adopted transform strategies: (1) direct fusion of original RGB and IR modalities [50,51,52] or via other transform domains (such as HSV or YUV), (2) sparse representation-based methods [53], (3) multi-scale decomposition-based methods [54], (4) saliency-based methods [55], (5) subspace-based methods [56] and (6) hybrid and other methods. With the development of deep learning technologies, various deep learning-based methods have shown high performance and potential in image fusion, e.g., auto-encoder (AE), convolutional neural network (CNN) and generative adversarial network (GAN). Li et al. [57] developed multi-scale encoder–decoder architectures to extract features and used a two-stage training strategy for a residual fusion network superseding handcrafted fusion strategies. Liu et al. [58] firstly generated saliency masks and the reconstructed images, then extracted salient features from the reconstructed double-input network as the weights to fuse them with reconstructed inputs to generate fusion results. Ma et al. [59] proposed an end-to-end generative adversarial network, with the generator concatenating major infrared intensities and visible imaging and the discriminator forcing the generator to accommodate more textures to cater to existing visible images.

However, the aforementioned deep learning-based pixel-level fusion methods primarily focused on reconstructing blurred infrared images combined with visible images, neglecting fine-grained pixel-level information and multispectral complementary feature fusion, which are crucial for high-resolution airborne multispectral images, thus leading to a shortage in fine details of generated images. Moreover, existing pixel-level fusion methods fail to bridge the large gaps in modality characteristics between visible and infrared images. They also overlook accuracy-driven designs for object detection tasks. As a result, their abilities to fully exploit complementary information based on modality similarities are limited.

## 3. Materials and Methods

### 3.1. Overall Architecture

To compensate for the weakness of the pixel-level fusion module in high-level feature extraction, we utilize YOLO models as the base framework due to their superior feature extraction capabilities, particularly provided by the CSP series modules (e.g., Bottleneck, C3, C2F). Specifically, we adopt YOLOv10s as the base architecture for its exceptional efficiency–accuracy balance. Compared to the previous YOLOv8 model, YOLOv10 eliminates NMS through its dual-label assignment strategy: the One-to-Many head enriches supervisory signals during training by generating the top-10 multiple predictions per ground truth, serving as an auxiliary assignment, while the One-to-One head assigns exactly one prediction, which is exclusively applied in detection inference. Though dual-label assignment may increase training time consumption, it results in a 65% decrease in inference latency due to its NMS-Free strategy, as only the One-to-One head is required during inference. YOLOv10 ensures its inference accuracy by applying a consistent matching metric to leverage the gap between two assignments. Other holistic efficiency–accuracy driven blocks, like SCDown, C2FCIB, SPPF and PSA, are incorporated to further enhance both inference speed and accuracy.

The overall architecture of our proposed FMPFNet is shown in Figure 2. The MLWF module is introduced to replace the first CBS feature extraction layer, which makes our YOLO-based model better at extracting features from the original multispectral images. The YUV color transformation strategy is applied in the MLWF module to moderate spectral imbalance between both modalities and reduce computation costs. Furthermore, the DMDTM training strategy is proposed to overcome the imbalance in visible–infrared training to improve detection performance for low-light illumination images. Additionally, the yolov10 OBB (oriented bounding box) head is designed, which is capable of both NMS-Free training and rotated object detection, then NWD loss is adopted in the model to improve small object detection performance.

### 3.2. MLWF Module

Pixelwise operators like concatenation and summation are widely used as pixelwise algorithm operations for they can utilize the spatial similarity of different modalities, while multiplication is usually applied in feature attention. However, these operations cannot determine whether features from different modalities are reliable, indicating that simply applying them may lead to interference between both modalities when at least one of the modalities contains ambiguous information. To this end, selection operators like Softmax are usually applied to allow the better component to dominate. However, a simple combination of summation (or concatenation) with selection operators will weaken the unique inherent features of visible and infrared images and neglect channel differences that significantly affect fusion results.

Given the drawbacks of simple pixelwise operations, we proposed MLWF, which contains a two-tier cascade modality reconstruction sub-module followed by a feature fusion sub-module, as shown in Figure 3. MLWF firstly generates attention maps that highlight the spectral and textural features of the original modalities within the modality reconstruction sub-module, which consists of a selection stage and an attention stage. The selection stage evaluates the light intensity of both modalities and assigns pixelwise importance weights using a Softmax operator. The attention stage generates feature maps based on individual modality features and the weight information from the selection stage. By applying the selection stage and the attention stage, the attention map simultaneously achieves redundant feature selection and unique feature enhancement. In the fusion sub-module, concatenation and CBS are applied to the reconstructed multispectral images with attention information, followed by the squeeze and excitation block (SE), which further enhances attention and complementary feature fusion.

The detailed structure of the modality reconstruction sub-module is illustrated in Figure 4. At the beginning of the selection stage, color space transformation (CST) is applied in the visible branch. The color space of visible images is transformed from RGB to YUV. Then, the luminance component *Y* instead of the chrominance components (*U*, *V*) is applied along with the infrared input to generate the selection map. The output selection map can enhance the more effective structural features among both input modalities. The selection map generation is defined as follows: (1)selectionmap=sigmoid(conv(softmax(α×Y,β×IR)))
where α and β are normalization parameters that balance the light intensity scales between both modalities, Softmax is a pixelwise operator that selects the more effective modality depending on illumination, conv denotes 1 × 1 convolution and the sigmoid activator smooths and normalizes the output of the selection map.

The mask maps of the respective modalities are generated by multiplying the selection map with the light intensity value of both modalities (*Y* component of visible light along with infrared) at the end of the selection stage: (2)$$\begin{array}{c} maskRGB=\mathrm{CSIT}(Y⊙selection\;map,U,V)\\ maskIR=IR⊙selection\;map \end{array}$$
where CSIT stands for color space inverse transformation, which converts YUV back to RGB, and the ⊙ operator denotes element-wise multiplication. In fact, the mask map of the visible modality can be simply calculated within two pixel-wise addition operations given the result of the selection map, which can be expressed as follows, as will be proved feasible in Section 5.1: (3)maskRGB=RGB+(Y⊙selectionmap−Y)

The mask maps contain a combination of the unimodal input and the complementary features of both modalities, serving as the demarcation of the selection stage and attention stage. The 1 × 1 convolution and Sigmoid reveal unique features of different modalities. At the beginning of attention stage, the attention map is generated via a multiplication of the two aforementioned blocks, simultaneously enhancing inherent unimodal features and selecting information from complementary features, which can be expressed as follows: (4)$$\begin{array}{c} attnRGB=\mathrm{sigmoid}(\mathrm{conv}(R,G,B))⊙maskRGB\\ attnIR=\mathrm{sigmoid}(\mathrm{conv}(IR))⊙maskIR \end{array}$$

The reconstructed multispectral images integrate complementary selection and original feature attention enhancement through the following: (5)$$\begin{array}{c} outRGB=RGB⊙attnRGB+RGB\\ outIR=IR⊙attnIR+IR \end{array}$$

These reconstructed outputs are concatenated with multispectral images, then processed through the CBS block as well as the SE block for further feature fusion of the feature maps. Finally, the output of MLWF model is expressed as follows: (6)output=SE(CBS(concat(outRGB,outIR)))

### 3.3. Dynamic Modality Dropout and Threshold Masking

Though multispectral fusion models generally exhibit more satisfactory performance compared to unimodal models, an exception arises when it comes to extreme scenarios. Especially in dark scenes where visible images are almost unusable while infrared images remain slightly affected, infrared-only models remain stable whereas multispectral fusion models sometimes fail. This phenomenon shows that the performance of multispectral detectors is highly sensitive to visible signals in low-light conditions, indicating that they tend to over-rely on visible images due to inherent modality imbalance: visible images contain more channels and possess richer textural features compared to infrared images.

Instead of introducing additional modules to enhance infrared feature extraction, applying data augmentation is more suitable for real-time aerial detection. Typical methods including flips, rotations, scaling and color transforms are capable of increasing dataset variations by extending the training dataset [60,61], though they were originally designed for RGB images. To mitigate modality imbalance and force the model to concentrate more on the infrared modality, we propose a simple yet effective adaptive augmentation strategy during training consisting of two specialized designs called Dynamic Modality Dropout and Threshold Masking (DMDTM).

Multispectral detectors usually tend to overlook the infrared modality when dealing with moderate lighting, leading to inadequate training on infrared images. To tackle this issue, random dropout of visible images by randomly replacing the chosen visible image with global average values is proposed to force the model to learn solely from infrared images and increase infrared concentration in daylight scenes. Given the process of training, the dropout rate is defined as follows: (7)$$dropout\;rate=\left\{\begin{array}{cc} 0, & \mathrm{when}\;\frac{e}{E}<0.2\;\mathrm{or}\;\frac{e}{E}>0.8\\ 1.25b_{0}-1.5625b_{0}\frac{e}{E}, & \mathrm{when}\;0.2<\frac{e}{E}<0.8 \end{array}\right.$$
where *e* is the current training epoch and *E* is the total training epoch. We denote $b_{0}$ as the initial dropout rate at the beginning of when random dropout is activated, which is set to 0.24. Throughout the process of training, the dropout rate decreases smoothly. Random dropout is activated neither at the beginning nor at the end of training to avoid training settings diverging from original datasets. Through this strategy, both the quantity ratio and diversity of infrared scenes increase, moderating the risk of inadequate training concerning ignored scenes.

To fully exploit infrared modality information, an adaptive luminance masking strategy is applied on visible images to prevent the multispectral visible branch from learning faint features or even noises in dark illuminations. By delaying the model from learning from visible signals in low-luminance areas, the infrared branch of model will be activated at the beginning of training, while valuable visible information will also emerge as training progresses. Specifically, instead of a fixed threshold with decreasing rate strategy that only increases the ratio of low-light scenes, a decreasing threshold strategy is adopted since it can both overcome visible-light failure in dark scenes and preserve complementary feature information in dim light.

On the one hand, according to statistics regarding pixels with luminance below 60 (above which regions contains adequate information) for low-light scenes in the Drone Vehicle dataset, as shown in Figure 5, the initial luminance threshold value γ is set to 40. This value corresponds to the pixel density convergence point in low-light scenes, demarcating the boundary between dark scenes and street lighting. On the other hand, visible-light detectors primarily rely on gradient differences between adjacent pixels, indicating that higher luminance pixels are more informative in low-light scenes. To adaptively balance luminance significance and spatial prevalence, we combine them to design the time coefficient of training using a dynamic threshold mask: (8)$$\delta_{n}=n^{m}\sigma_{n}$$
where *n* denotes the luminance value, $\delta_{n}$ represents the time weight of a certain luminance value proportional to training epochs, $\sigma_{n}$ denotes the proportion of pixels corresponding to a certain luminance value as shown in Figure 5, and m adjusts the balance between luminance importance and distribution, which is set to 0.5 in our experiments because in dark scenes, street lights significantly improve local luminance and hinder attention on uncovered areas, with m less than 1 improving the importance of dark pixels. According to the aforementioned discussions, the masking threshold of certain training epoch can be expressed as follows: (9)$$\mathrm{threshold}=n,\;\mathrm{when}\;\frac{\sum_{i=n+1}^{40}\delta_{i}}{\sum_{i=0}^{40}\delta_{i}}<\frac{e}{E}<\frac{\sum_{i=n}^{40}\delta_{i}}{\sum_{i=0}^{40}\delta_{i}}$$

Throughout the process of training, the luminance threshold of visible images smoothly decreases, enabling the model to dynamically determine whether to prioritize complementary feature fusion or rely predominantly on infrared features according to various low-light circumstances.

### 3.4. Aerial Object Detection Specialized Designs

The YOLOv10 model shows satisfactory performance and a faster inference speed through its NMS-free training strategy but lacks support for arbitrary direction object detection. To solve this problem, inspired by the previous YOLOv8 OBB head, we propose a YOLOv10 OBB head that decouples the angle prediction from bounding box regression and classification. While bounding box regression and classification tasks in YOLOv10 are decoupled, they still rely on similar multiscale feature information. In contrast, the angle prediction for oriented bounding boxes requires only texture orientation information from the feature maps. Our proposed model calculates the angle once before dual-label assignments rather than separately calculating it in the one-to-one head and the one-to-many head. Depthwise separable convolution composed of a 3 × 3 convolution with groups equal to the input channel followed by a 1 × 1 convolution is applied in the angle regression. This design is similar to that used in classification prediction for these light-weighted convolution designs, which does not affect performance greatly.

Along with the OBB head design, we also propose an adjusted training loss function adapted from the YOLOv8 OBB loss used for aerial small-object detection in the YOLOv10 OBB model. The original loss function consists of three components, i.e., ProbIoU loss as bounding box loss, DFL (distribution focal loss) and cross entropy loss applied in classification loss. ProbIoU loss and DFL loss exhibit advantages in oriented bounding box object detection by evaluating the possible distributions of target bounding boxes rather than certain values. However, they still exhibit high sensitivity to position deviations in small objects like other loss functions. To this end, we introduce NWD loss, which calculates Wasserstein distance based on optimal transport theory and normalized Gaussian distribution assumptions. NWD loss enjoys scale invariance and smoothness when it comes to location deviations, making it more suitable for small -bject detection. Since ProbIoU loss calculates the Bhattacharyya Coefficient, concentrating on the evaluation of oriented bounding boxes, we partly replace ProbIoU loss with NWD loss to calculate bounding box loss. Our proposed loss function is expressed as follows: (10)$$L_{total}=c_{1}\left[\lambda L_{Prob}+\left(1-\lambda \right)L_{nwd}\right]+c_{2}L_{cls}+c_{3}L_{dfl}$$
where λ balances ProbIoU (orientation accuracy) and NWD (small object robustness), which is set to 0.5 in our model. The coefficients $c_{1}$, $c_{2}$ and $c_{3}$ remain consistent with the baseline to preserve training stability.

## 4. Results

### 4.1. Experimental Settings

All experiments were implemented in PyTorch 3.10 on a Linux system equipped with an Intel Xeon Gold 6240C CPU and NVIDIA RTX 3090 GPU. The SGD optimizer was adopted with a momentum of 0.937 and weight decay of 0.0005. The initial learning rate was 0.01 and the learning rate of final epoch was 1 × $10^{-4}$. All of the original backbone of YOLO models were accessed on ultralytics [62].

### 4.2. Dataset and Metrics

The DroneVehicle [63] dataset is the largest full-time aerial RGB–Infrared cross-modal object detection dataset, comprising 28,439 pairs of images across 5 different vehicle categories: car, truck, bus, van and freight car. The dataset is divided into a training set with 17,990 image pairs, a validation set with 1469 image pairs and a test set with 8980 image pairs. These images cover diverse illumination conditions, including daylight, dim and dark scenes. Various weather conditions like misc and thick fog are also included. Rotated annotations are provided for aerial or remote sensing arbitrary-direction object detection. Visible images and infrared images are separately annotated with VOC polygons, with infrared images containing more labels due to the fact that only infrared images remain available to human eyes in dark scenes (night). In order to evaluate our multispectral object detection model, we integrate these annotations in visible and infrared images. Specifically, infrared annotations are exclusively adopted in dark illumination scenes or image patches. The integrated annotations include 431,519 cars, 26,082 trucks, 16,692 buses, 12,866 vans and 17,200 freight cars in total. For daylight scenes, visible annotations are adopted in the original YOLO data augmentation training, and when DMDTM is applied in training, only infrared annotations are adopted for visible-masked images. The original size of each visible and infrared image in the dataset is 840 × 712 pixels, with 100 pixels of blank padding around for the purposes of oriented bounding box annotation; thus, the size of each image is 640 × 512 pixels. The visualized statistics of the DroneVehicle dataset are shown in Figure 6.

In our experiments, we applied a series of metrics for object detection assessment, i.e., GFLOPs (Giga floating point of operations), FPS (frames per second) and mAP50 (mean average precision for IoU = 0.50), indicating model size, inference speed and detection performance, respectively. The performance metrics of evaluated models can be calculated as follows: (11)$$\mathrm{Precision}=\frac{TP}{TP+FP}$$(12)$$\mathrm{Recall}=\frac{TP}{TP+FN}$$(13)$$AP=\int_{0}^{1}\mathrm{Precision}\left(\mathrm{Recall}\right)\;d\left(\mathrm{Recall}\right)$$(14)$$mAP=\frac{1}{n}\sum_{i=1}^{n}AP_{i}$$
where TP stands for True Positive, which is a correctly predicted object of a certain class; FP stands for False Positive, which is a sample incorrectly predicted to belong to a certain class; and FN stands for False Negative, which is an object of a certain class that is not successfully predicted. AP represents the Average of Precision across all recall levels and mAP50 is the mean Average of Precision of every class for which samples are classified correctly and regressed with an IOU between the sample and ground truth greater than 0.5 with respect to true positives.

### 4.3. Ablation Studies

#### 4.3.1. Effectiveness of YUV Color Space Transformation

As shown in Section 3.2, the YUV color space transformation is applied during the selection stage to cope with multispectral channel differences. Apart from the YUV color transformation, the HSV color transformation is also widely used in computer vision. Other typical transformations like CMYK, YCbCr and HSI, though distinct in expression, are functionally equivalent in specific expressions to the RGB, YUV and HSV transformations, respectively, in our framework. What’s more, to verify the effectiveness of our color transformation strategy, the modality selection strategy of simply concatenating the IR images and original visible image followed by a 1 × 1 convolution is also applied in the following ablation experiment. The base model removes the selection stage compared to our proposed model without DMDTM data augmentation usage. All model parameters except those in the selection stage remain constant. The results of the experiments are shown in Table 1.

Ablation studies confirm the superiority of the YUV transformation over other selection strategies. Simply concatenating the original images increases model complexity while helping little with overall performance. The HSV color transformation method shows comparable accuracy with suboptimal inference speed. In contrast, the YUV color transformation enjoys a 10% overall speed advancement due to its linear matrix transformation, with way simpler operations compared to the HSV color transformation.

#### 4.3.2. Effectiveness of MLWF Sub-Modules

In order to verify the necessity of both stages in our pixel-level fusion module, MLWF, ablation experiments were performed on the Drone-Vehicle dataset to evaluate the YOLOv10s-OBB baseline model, an selection stage-only model, an attention stage-only model and the model containing both. Furthermore, the significance of the SE block is also verified. The baseline model utilizes the original YOLOv10 network, and identical parameters are applied for each model training using our proposed OBB head and NWD-ProbIoU box loss. The results of the experiments are shown in Table 2, where ‘✓’ denotes the inclusion of MLWF components.

FMPFNet containing all proposed components achieves an mAP of 76.19%, improving performance by 1.77% compared to the baseline model. We note that only applying the selection stage without the attention stage slightly degrades overall performance, indicating that modality selection cannot be applied alone as it overlooks the complementary feature combination and unique feature enhancement, which are crucial for multispectral object detection. Only applying the attention stage yields a performance improvement of 0.96%, lower than that of applying both stages (1.25%), which shows that modality selection prior to attention would boost the overall performance, indicating the necessity of complementary feature selection. SE removal degrades the performance by 0.52% when both stages are applied, which shows that the SE block followed by CBS would further strengthen multispectral feature fusion. This study demonstrates that although the selection stage, attention stage and SE blocks in FMPFNet introduce modest computational overhead and reduce inference speed by 10.2%, they significantly enhance model accuracy. Our model maintains a lightweight model size and real-time inference capability, achieving an optimal speed–accuracy trade-off.

#### 4.3.3. Effectiveness of DMDTM Augmentation Strategy

Figure 7 shows three representative examples labeled as (1)–(3), with their characteristics and ground truth bounding boxes visualized in (a)–(d). As described in Section 3.3, the default YOLO augmentation methods fail to address the modality imbalance between visible and infrared inputs, leading to inadequate utilization of infrared features, thus yielding suboptimal detection accuracy, as shown in Figure 7e,f. Our proposed DMDTM augmentation strategy is applied before original YOLO augmentation, which strengthens the advantages in low-light conditions and compensates for the ignorance in well-lit scenes of the infrared modality. The performance results of the different methods are shown in Figure 7, where (e) and (g) show the effectiveness of DMDTM.

We evenly split the train and test dataset based on lighting conditions into three categories, i.e., daylight, dim and dark, which will be evaluated separately in the following ablation studies of the augmentation strategies. To validate the effectiveness of the DMDTM method, we compare it with two strategies also aimed at reducing spectral imbalance: random oversampling, which randomly copies low-light (dim/dark) samples, and SMOTE [64], which synthesizes original low-light images to generate new low-light samples. To further increase the ratio of low-light scenes without generating a training burden, we meanwhile randomly under-sample light scene images. Both comparison methods adjust the original light–dim–dark ratio from 1:1:1 to 1:2:3 and 1:3:5 using the python package imblearn [65]. Based on our proposed FMPFNet, the results of experiments on different augmentation strategies are shown in Table 3.

By only increasing the ratio of low-light scenes, random oversampling and SMOTE fail to effectively improve performance in low-light scenes. Their overall performances degrade when the quantity ratio of low-light scenes is too high due to sharp performance decline when it comes to daylight scenes. In contrast, our proposed model strategy achieves significant improvement (2.29% increase) in dark scenes with little sacrifice (0.39% decrease) in daylight scenes. The result indicates that our DMDTM successfully forces the detection model to draw more attention to infrared images in dark scenes instead of over-relying on the visible modality, thus improving overall performance.

### 4.4. Comparisons

In this study, we evaluate our proposed pixel-level fusion model FMPFNet on the DroneVehicle dataset and benchmark it against representative models and state-of-the-art feature-level fusion models. We compare it to the up-to-date YOLOv8, YOLOv10 and YOLOv11 models given their similar structure. YOLOv8 is widely adopted with its fine-designed OBB object detection, YOLOv10 is optimized for lightweight deployment and higher inference speed and YOLOv11 represents the latest advancements. Besides the original YOLO series, we also include YOLO fusion, which enhances model-specific features and model-shared feature selection for remote sensing multispectral small object detection. As for multispectral models, UA-CMDet [63] is applied since it was co-designed with the DroneVehicle dataset. Typical multimodel detectors included have been validated as effective on widely used RGB-IR datasets like FLIR and VAIST, while the latest feature-level fusion models exhibit higher accuracy with larger model size. Last but not least, we apply the DMDTM data augmentation strategy on our FMPFNet, and a comparison experiment without DMDTM on our FMPFNet is also included. The comparison results of different methods are shown in Table 4.

Results of comparisons among YOLO-series models show that YOLOv10 outperforms YOLOv8 and YOLOv11 not only in inference speed but also in accuracy, which highlights its comprehensive superiority in aerial multispectral object detection and confirms YOLOv10 as the optimal base framework for our proposed FMPFNet. The comparisons between multispectral models demonstrate that FMPFNet with the DMDTM augmentation strategy exhibits competitive accuracy with significantly advantages over state-of-the-art models in model size and inference speed. Among models of similar size, FMPFNet prevails over other prevalent YOLO-based methods in multispectral object detection performance, beating YOLOv8s by 3.46%, YOLOv10s by 2.38% and YOLOFusion by 2.26%. In comparison with state-of-the-art feature-level fusion methods, our FMPFNet surpasses the majority of methods by more than 2%, CALNet by 1.4% and FMCFNet by 0.05%, meanwhile exhibiting outstanding advantages in model size and inference speed. Even without the DMDTM strategy, our proposed FMPFNet still exhibits satisfactory performance, beating the majority models in all aspects. In conclusion, FMPFNet exhibits state-of-the-art performance with superiority in real-time multispectral object detection with limited resources, and DMDTM further improves its performance without sacrificing inference speed.

## 5. Discussion

### 5.1. Effectiveness Interpretation of YUV Transformation

As introduced in Section 3.2, the YUV transformation method has been introduced as the color space transformation module. The transformation formula from RGB to YUV color space is(15)$$\left[\begin{array}{c} Y\\ U\\ V \end{array}\right]=\left[\begin{array}{ccc} 0.299 & 0.587 & 0.114\\ -0.147 & -0.289 & 0.436\\ 0.615 & -0.515 & -0.1 \end{array}\right]{\left[\begin{array}{ccc} R & G & B \end{array}\right]}^{T}$$

The infrared modality contains a single channel that reflects the thermal radiation intensity, while the *Y* component contains luminance information. They share similarities in encoding the structural features of the respective modalities, enabling the model to better contrast and select from redundant and conflicting features of the visible and infrared modalities. According to Section 3.2, the mask map of the visible modality can be summarized as follows: (16)$$maskRGB=\mathrm{CSIT}({\mathrm{CST}}_{Y}\left(R,G,B\right)⊙selection\;map,U,V)$$

The selection map, determined by the *Y* component and infrared input, is not directly relevant to the visible chrominance components. One of the principal advantages of the YUV model in image processing is the decoupling of luminance and color information, ensuring operations on one of them will not interfere with the other. The *Y* component (luminance) contains rich illumination-related information. By integrating it with the selection map, the model successfully focuses on structural features in daylight scenes without altering visible chrominance components. With a decoupled relationship between the *Y*, *U* and *V* components, the YUV transformation strategy eliminates the need for calculating color difference components, reducing computation cost during training and inference.

For further discussion, we use RGB to represent the original visible images, which have three channels. The mask map of the visible modality is generated by operations on the *Y* component of the original visible image input. The following expressions, simultaneously containing the visible image representation RGB and luminance component *Y*, indicate that calculations containing *Y* are broadcast to match the three-channel RGB. From the perspective of differential equations, the mask map of the visible modality can be expressed as follows:(17)$$maskRGB=RGB+\frac{\partial S}{\partial Y}dY+\frac{\partial S}{\partial U}dU+\frac{\partial S}{\partial V}dV=RGB+\left(Y^{\prime }-Y\right){\left[\frac{\partial R}{\partial Y}\frac{\partial G}{\partial Y}\frac{\partial B}{\partial Y}\right]}^{T}$$

The inverse transformation that converts the color space from YUV to RGB is as follows: (18)$$\begin{array}{cc} R & =Y+1.14V\\ G & =Y-0.394U-0.581V\\ B & =Y+2.032U \end{array}$$

The selection map is determined by the *Y* component input value and remains constant even when the *Y* component changes during the process, which means it will not be calculated during backpropagation. Based on aforementioned discussions, it is easy to find that(19)$$\frac{\partial R}{\partial Y}=\frac{\partial G}{\partial Y}=\frac{\partial B}{\partial Y}=1$$

So, we can conclude that(20)$$maskRGB=RGB+\left(Y^{\prime }-Y\right)=RGB+\left(Y⊙selection\;map-Y\right)$$

The output mask map of the visible modality can be calculated within two simple pixel-wise addition operations once the selection map is calculated. The YUV transformation method further reduces computation overhead and minimizes inference speed declination compared to other color transformations.

### 5.2. Parameter Ablation and Analysis in DMDTM

The initial dropout rate $b_{0}$ and initial threshold value γ in DMDTM constitute critical hyperparameters. Parameter sensitivity experiments on these parameters are conducted to enhance performance of our FMPFNet model, with experimental results shown in Figure 8 and Figure 9.

Ablation studies reveal that the optimal value of $b_{0}$ most probably ranges between 0.24 and 0.32. Performance results tend to decline steadily when $b_{0}$ is set to 0.40 or more, which indicates that overemphasizing the importance of the infrared modality can negatively impact overall performance, which is also proved in Section 4.3.3. When it comes to γ, results indicate 40 as near-optimal. No significant performance improvements occur when γ is too low. By contrast, accuracy increases steadily when γ falls in the interval [10, 20), which aligns with the predominant noise distribution range in dark scenarios of the DroneVehicle dataset, demonstrating that the improvement in the multispectral model’s robustness against visible noises in low-light conditions substantially benefits overall performance. When $b_{0}$ is set to over 40, the accuracy no longer increases, as very few pixels in dark scenes have luminance values above 40. It should be emphasized that these two hyperparameters (especially γ) are closely tied to the characteristics of the dataset, so the parameters determined in our experiments are specific to the DroneVehicle dataset. For other drone-based multispectral object detection datasets, comprehensive parameter re-optimization is strongly recommended.

### 5.3. Limitations and Future Work

We explored pixel-level fusion methods originally for their advantages regarding model size and inference speed over feature-level fusion, which is friendly to aerial object detection in airborne platforms that require real-time detection with limited resources. The MLWF module is plug-and-play and can be embedded in the first stage of CNN-based detectors, but it lacks multiscale feature awareness, thereby relying on detectors exhibiting excellent abilities in extracting multiscale multispectral features such as YOLO models with their specialized feature extraction CSP module. The DMDTM strategy has proved effective in pixel-level fusion methods for its pixelwise threshold dropout operation, with advantages concerning no additional model complexity and computation cost. However, it exhibits suboptimal performance in feature-level fusion methods, which is probably due to conflicts between pixelwise operations and high-level feature fusion, a result of changing the textual feature information of visible feature maps then interfering with subsequent fusion.

We will improve the generalization of our proposed methods in future work. Specifically, we plan to explore multiscale map generation directly within our pixel-level fusion module, such as applying multiscale max-pooling on original inputs followed by attention-based designs, which may improve performance without introducing much computation cost. Meanwhile, we will further investigate the modality differences between multispectral inputs to develop more systematic approaches for further boosting infrared feature exploitation in low-light conditions without compromising daylight scene performance.

## 6. Conclusions

This paper presents a robust framework for real-time airborne multispectral object detection through pixel-level fusion and dynamic modality augmentation. The proposed MLWF module addresses modality imbalance by applying a cascade attention-based modality reconstruction module integrating a YUV color transformation, effectively reducing computational costs while adequately utilizing feature complementarity and dealing with redundancy. The DMDTM strategy further improves detection ability in low-light scenes by adaptively balancing attention between RGB and IR modalities, successfully improving overall performance. The OBB dual-label assignment design enables rotated object detection via the YOLOv10 model and the proposed loss function optimization improves small-object detection accuracy. Through extensive experiments on the DroneVehicle dataset, our proposed model and augmentation strategy prove to achieve state-of-the-art performance for real-time aerial object detection. Considering the model complexity and computation cost, there is enough room for our model to be further deployed in airborne resource-limited platforms for both full-time and real-time aerial image processing, enabling 24/7 monitoring and response and adequately utilizing the flexibility advantages of drones and satellites. 

## Figures and Tables

**Figure 1 sensors-25-03039-f001:**
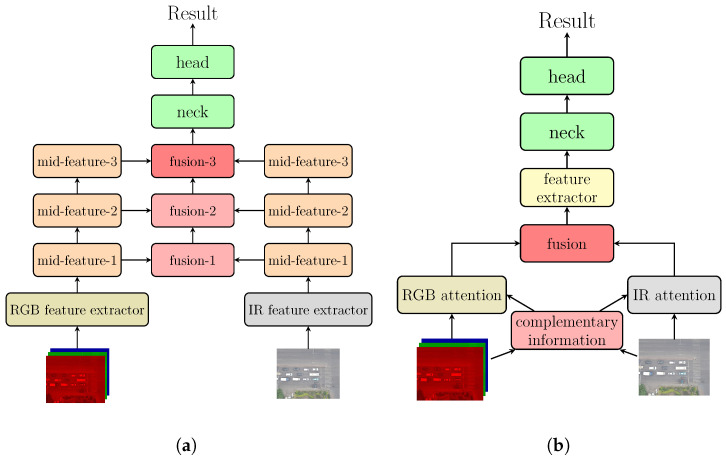
Fusion methods based on different stages. (**a**) Typical feature-level fusion. (**b**) Attention-based pixel-level fusion.

**Figure 2 sensors-25-03039-f002:**
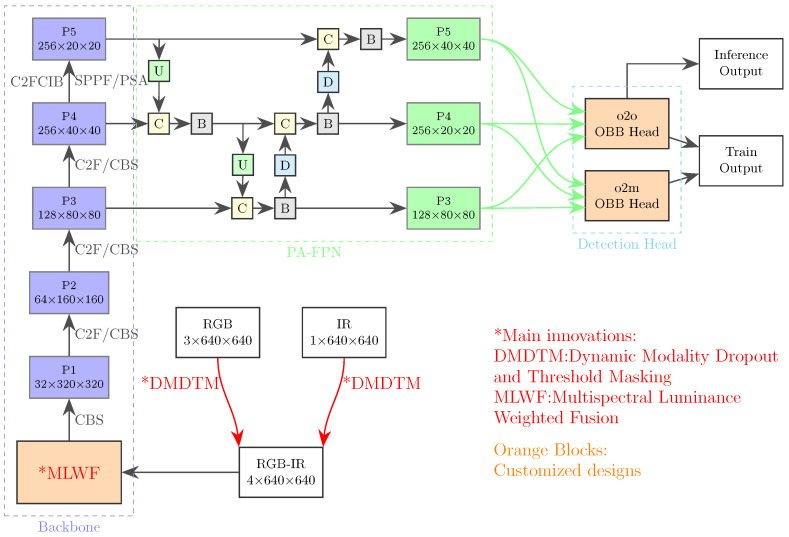
Overall architecture of FMPFNet. YOLOv10s is applied as the base model. The DMDTM augmentation strategy is applied before model training to moderate multispectral modality imbalance. The CMWF pixel-level fusion module replaces the first feature extraction layer of the backbone. The oriented bounding box dual-label assignment head is designed for aerial object detection.

**Figure 3 sensors-25-03039-f003:**
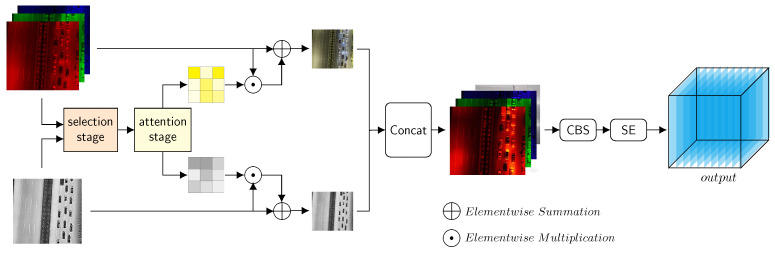
Structure of the MLWF module. Firstly, the features of original inputs are enhanced by the modality reconstruction sub-module during the selection stage and attention stage. Then, reconstructed images are concatenated and further fused in the feature fusion sub-module.

**Figure 4 sensors-25-03039-f004:**
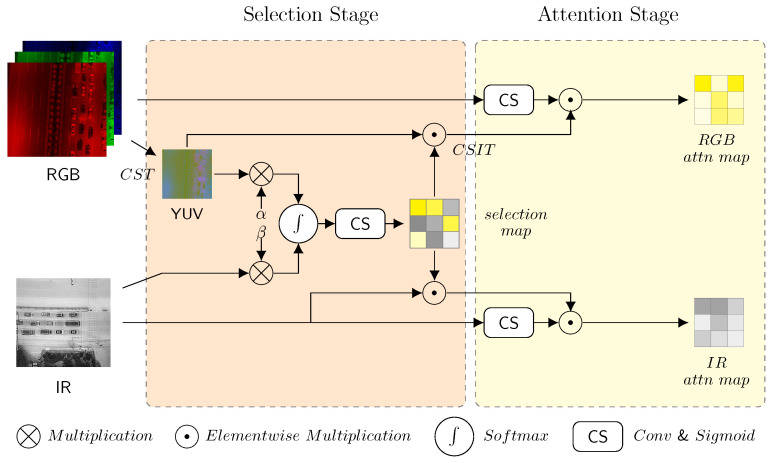
The architecture of the proposed RGB-IR modality reconstruction sub-module. The selection stage generates the selection map to analyse the importance of different modalities based on light intensity. The attention stage combines information from unimodal characteristics and the selection map to better enhance feature representations deriving from the respective modalities.

**Figure 5 sensors-25-03039-f005:**
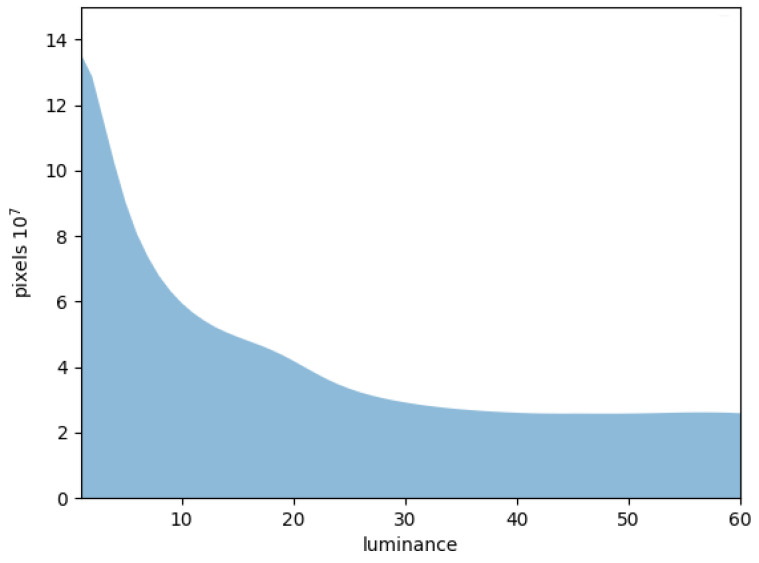
Statistics concerning pixels in DroneVehicle dataset with visible modal luminance below 60.

**Figure 6 sensors-25-03039-f006:**
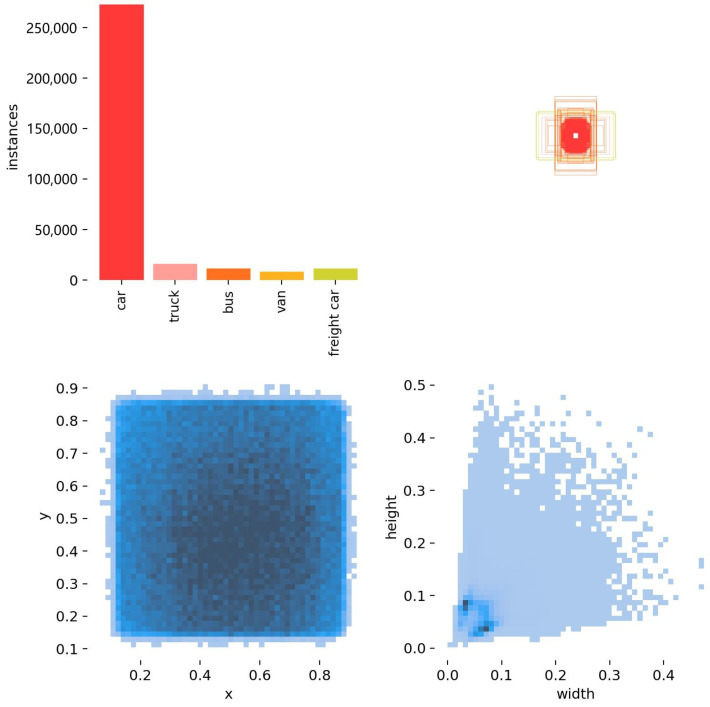
Visualization of training data distribution of the DroneVehicle dataset. The top-left sub-figure contains instance classification statistics. The top-right sub-figure contains the distribution of bounding box aspect ratios. The bottom-left sub-figure contains the normalized location distribution of ground truths. The bottom-right sub-figure contains the width and height distribution of ground truths with normalized dimensions.

**Figure 7 sensors-25-03039-f007:**
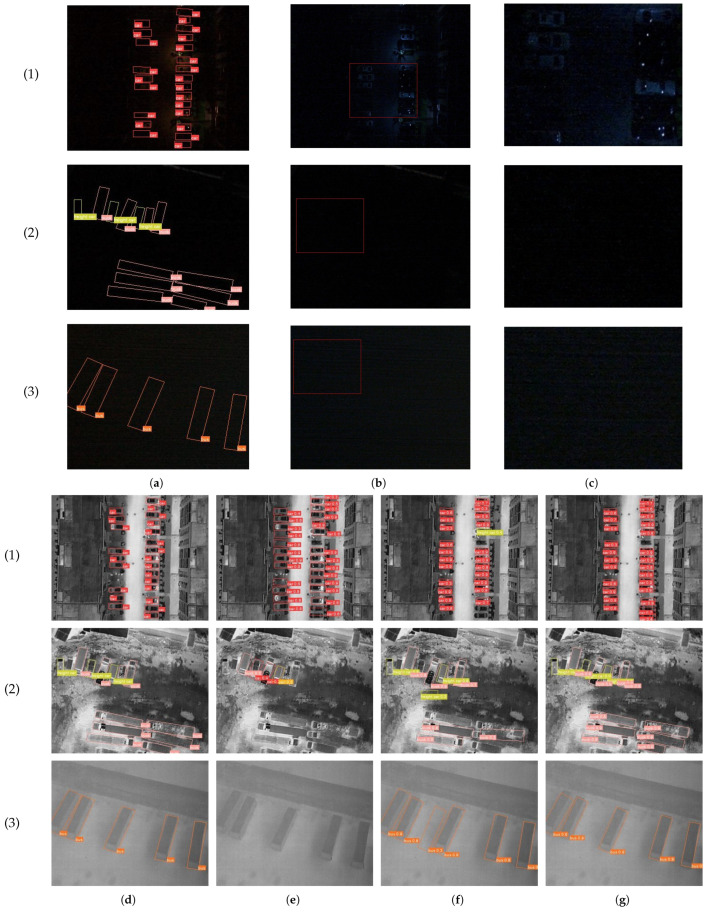
Representative examples of different methods in low-light conditions. From top to bottom are three examples with visible images containing (1) slight features, (2) no features and (3) noises. From left to right are different situations: (**a**) ground truth of visible modality, (**b**) original images with areas of interest given by red rectangles, (**c**) enlarged view of areas of interest, (**d**) ground truth of infrared modality, (**e**) detection results of multimodal fusion model, (**f**) detection results of infrared-only model and (**g**) detection results of multimodal fusion model with our proposed augmentation strategy.

**Figure 8 sensors-25-03039-f008:**
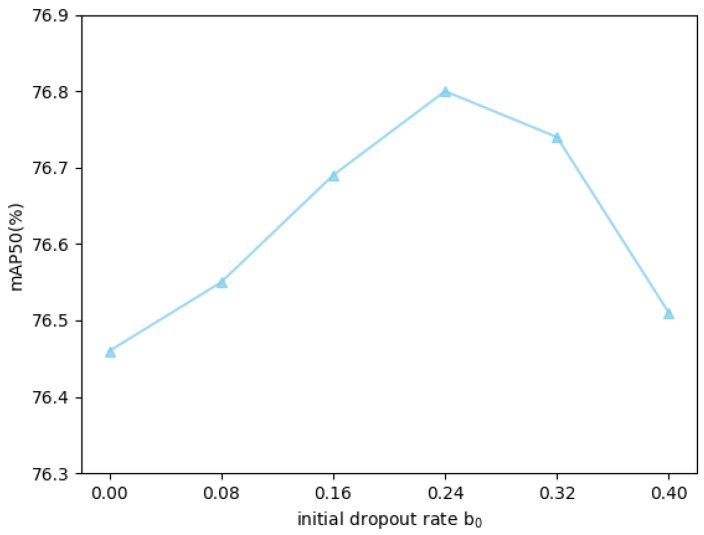
Parameter ablation of initial dropout rate $b_{0}$.

**Figure 9 sensors-25-03039-f009:**
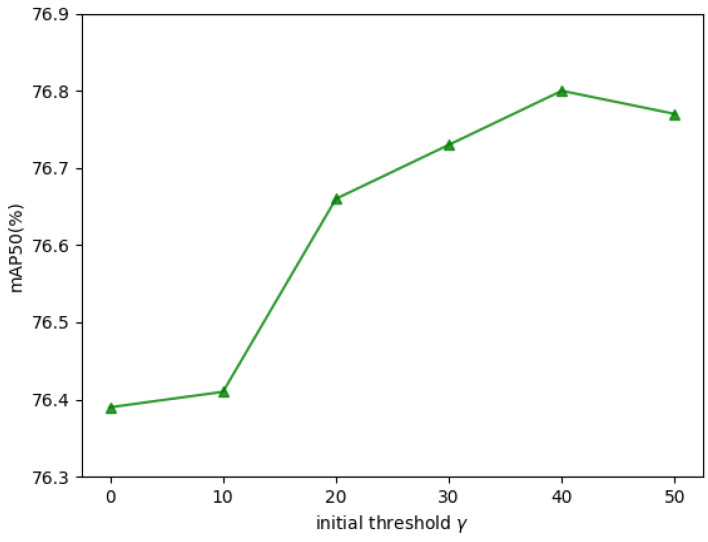
Parameter ablation of initial dropout rate γ.

**Table 1 sensors-25-03039-t001:** Evaluation of different selection design methods of our model. RGB represents the method that concatenates both modalities then applies selection via a 1 × 1 convolution. YUV and HSV represent the respective color transformations with our proposed selection stage.

Methods	FPS	mAP
RGB	126	75.64
YUV (ours)	132	76.19
HSV	115	76.02

**Table 2 sensors-25-03039-t002:** Evaluation of different components in our proposed MLWF module, where SS stands for selection stage, AS stands for attention stage and SE stands for squeeze and excitation block.

Methods	SS	AS	SE	GFLOPs	FPS	mAP
baseline				26.212	147	74.42
1	✓			26.218	141	74.35
2		✓		26.220	139	75.36
3	✓	✓		26.226	134	75.67
4			✓	26.220	144	74.88
5	✓		✓	26.227	138	74.83
6		✓	✓	26.228	137	75.66
FMPFNet	✓	✓	✓	26.234	132	76.19

**Table 3 sensors-25-03039-t003:** Evaluation of different training strategies, where RO stands for random oversampling. The mAP50 is applied as the accuracy metric for all scenes. The best indicators are **bold**.

Detectors	Daylight	Dim Light	Dark	Overall
baseline	**82.03**	75.94	70.60	76.19
RO (1:2:3)	81.50	75.80	71.35	76.21
SMOTE (1:2:3)	81.63	75.74	71.39	76.25
RO (1:3:5)	80.32	75.64	71.76	75.91
SMOTE (1:3:5)	80.54	75.63	71.80	75.99
DMDTM (ours)	81.64	75.87	**72.88**	**76.80**

**Table 4 sensors-25-03039-t004:** Comparison of different models. The best indicators are **underlined and bold**, and the second best are **bold**.

Detectors	GFLOPs	FPS	Car	Trunk	Bus	Van	Freight Car	mAP
YOLOv8s (OBB)	29.5	103	94.61	70.37	92.86	54.18	54.68	73.34
YOLOv10s (OBB) ^1^	**26.2**	**147**	94.84	70.62	93.64	55.62	57.39	74.42
YOLOv11s (OBB)	**24.6**	130	94.77	71.83	93.07	53.94	56.03	73.93
UA-CMDet [63]	-	-	87.51	60.70	87.08	37.95	46.80	64.01
Halfway Fusion [38]	77.31	31	89.85	60.34	88.97	46.28	55.51	68.19
CIAN [66]	70.36	32	89.98	62.47	88.90	49.59	60.22	70.23
AR-CNN [67]	104.3	28	90.08	64.82	89.38	51.51	62.12	71.58
C2Former [68]	100.9	30	90.21	68.26	89.81	**58.47**	64.62	74.23
YOLO Fusion [19]	25.7	61	91.46	72.71	90.58	57.62	60.35	74.54
CAGTDet [23]	120.6	26	90.82	69.65	90.46	55.62	**66.28**	74.57
CALNet [69]	174.54	-	90.30	76.15	89.11	58.46	62.97	75.40
FMCFNet [4]	200.43	-	90.33	**77.71**	89.35	**60.71**	**65.65**	**76.75**
FMPFNet (Ours) ^2^	26.3	**132**	**95.23**	73.61	**94.45**	58.01	59.66	76.19
FMPFNet (Ours)	26.3	**132**	**95.18**	**76.79**	**94.46**	57.08	60.47	**76.80**

^1^ YOLOv10 utilizes our OBB head and its original backbone. ^2^ Our method without DMDTM augmentation.

## Data Availability

The DroneVehicle dataset is available at https://github.com/VisDrone/DroneVehicle (accessed on 29 December 2021).

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
