# Peer review of "Real-Time Aerial Multispectral Object Detection with Dynamic Modality-Balanced Pixel-Level Fusion"

_sensors, 2025, doi:10.3390/s25103039_

Round 1

Reviewer 1 Report

Comments and Suggestions for Authors

The paper proposes a real-time aerial multispectral object detection framework, named FMPFNet, combining RGB and infrared imagery through a pixel-level fusion method called Multispectral Luminance Weighted Fusion. It addresses modality imbalance by employing dynamic augmentation strategies, including Dynamic Modality Dropout and Threshold Masking, particularly improving detection performance in low-light scenarios. Experiments conducted on the DroneVehicle dataset demonstrate that the proposed method outperforms state-of-the-art approaches in terms of accuracy and inference efficiency. In general, this paper is well-written and very easy to follow. However, some weaknesses must be addressed before publication.

  1. Although the manuscript proposes the Dynamic Modality Dropout and Threshold Masking (DMDTM) strategy, detailed exploration on how different dropout ratios impact model performance across varying illumination scenarios is lacking. Moreover, sensitivity analysis regarding threshold selection is insufficient.
  2. The proposed method, while claiming to be real-time, integrates multiple modules (attention, selection, SE blocks, dual-label assignments, etc.), each of which may introduce incremental complexity. A clearer justification of each module’s necessity or redundancy, backed by detailed ablation studies, is needed.

  3. The manuscript contains some grammatical errors and awkward phrasings, e.g., “inherit” instead of “inherent”, “duel” instead of “dual”, “poste more challenges” instead of “pose more challenges”. Careful proofreading is required.

  4. Some works about RGB-T fusion should be cited in this paper to make this submission more comprehensive, such as 10.1109/TPAMI.2024.3511621, 10.1109/TPAMI.2024.3475472.

Author Response

Thank you very much for taking time to carefully review our article and we're sorry for the late reply. As far as we know, four improvement suggestions are listed in your review. We have carefully considered them and thus we have made some revisions in our article with critical revisions filled orange. In this letter, we will provide detailed response to your comments as follows: 

Comment 1: Although the manuscript proposes the Dynamic Modality Dropout and Threshold Masking (DMDTM) strategy, detailed exploration on how different dropout ratios impact model performance across varying illumination scenarios is lacking. Moreover, sensitivity analysis regarding threshold selection is insufficient.

Response 1: We agree that parameter ablation and analysis on parameter initial dropout rate and initial threshold value can make our model more comprehensive and convincing, so we add subsection 5.2 to describe this point.

Comment 2: The proposed method, while claiming to be real-time, integrates multiple modules (attention, selection, SE blocks, dual-label assignments, etc.), each of which may introduce incremental complexity. A clearer justification of each module’s necessity or redundancy, backed by detailed ablation studies, is needed.

Response 2: The application of our proposed attention stage and selection stage, along with addition of SE block would introduce little model complexity ,but they enable multispectral object detection capability and help notably improve inference accuracy, so applying them in our model is necessary. Dual-label assignments won't introduce any incremental complexity during inference because one-to-many head is only activated during training and not included during inference, according to YOLOv10. Our OBB head preserved this characteristic. We had already mentioned these points in the original article, but we have still further supplemented the explanations in subsections 3.1 and 4.3.2 of our revised article.

Comment 3: The manuscript contains some grammatical errors and awkward phrasings, e.g., “inherit” instead of “inherent”, “duel” instead of “dual”, “poste more challenges” instead of “pose more challenges”. Careful proofreading is required.

Response 3: We acknowledge that there are some mistakes in our article. The mentioned three mistakes are corrected. Additionally, a thorough proofreading is also conducted to the best of our ability.

Comment 4: Some works about RGB-T fusion should be cited in this paper to make this submission more comprehensive, such as 10.1109/TPAMI.2024.3511621, 10.1109/TPAMI.2024.3475472.

Response 4: We found that these works are specific and updated complicated yet effective feature-level fusion methods, so we applied them in subsection 2.2.

Reviewer 2 Report

Comments and Suggestions for Authors

Manuscript presents a novel, well-engineered framework for real-time aerial multispectral object detection using pixel-level fusion and dynamic modality balancing. The FMPFNet design, combining YUV-based luminance fusion and training-time modality dropout, is both innovative and practical for drone deployment scenarios.

However, to strengthen the manuscript, the authors are advised to work on the following issues:

  1. Language and Grammar Issues:
    1. Spacing errors: Missing/incorrect spacing before or after punctuation (references, periods, commas, brackets, listing numbers).
    2. Typos and word misuse:
      1. "propose" instead of "purpose" (Line 79)
      2. "then" instead of "them" (Line 153)
      3. "detctor" instead of "detector" (Line 286)
      4. "mode" instead of "model" (Line 218)
      5. Redundant: word "make" is repeated (Line 218)
      6. Word Figure is missing in front of figure number (Line 283)
      7. It should be "either" and "or" instead of "neither" and "nor" (Line 308-309)
      8. Line 368: "and" at the end of line is not required
      9. Line 384: instead of "only is", it should be "are"
      10. Line 460: it should be "under-sample" instead of "under-sampling"
  2. Incomplete or Ambiguous Sentences:
    1. Line 218-219: "extract features from the original multispectral image that..." unfinished clause.
    2. Line 281-282: Confusing phrasing; sentence lacks clarity.
    3. Line 347: Missing transition word disrupts sentence flow.
    4. Line 367: It should be "in PyTorch" instead of "on PyTorch".
  3. Visual and Figure Annotation Issues:
    1. Figure 4: Incorrect operation symbol (⊗ used instead of ⊙, text and equations defines and use the later symbol for elementwise multiplication).
    2. Figure 2: There should be elementwise multiplication operation and symbol ⊙ (as per equation (5)) instead of elementwise summation symbol for the part where attention map of each modality is multiplied with their respective modality.
    3. Figure 5, make it uppercase for first letter "statistics" in the caption.
    4. Figure 6: sub-figure numbering is missing for all four sub-figures. Also, caption doesn't explain the sub-figures. Missing explanation for "x" and "y" for bottom-left sub-figure. Missing annotation for top-right sub-figure.
    5. Figure 7: Sub-figures (d), (e), (f), and (g) are cluttered; adding vertical spacing and clearer layout would enhance the representation.
  4. Formatting and Typographical Errors:
    1. Numbers such as "28, 439" should be "28,439"; and "1469" should be "1,469" for consistency. Check for similar instances.
    2. Resolution values: "840*712"  should be "840×712". Check for similar instances.
    3. Equation citations: "of iou0.50" should be "for IoU = 0.5".
    4. "utilities" (Line 428) should be "utilize".
    5. Table citation error: "Table4 4" should be "Table 4".
    6. Misplacement of Equations 17 and 18  and related text.

With the above improvements, the manuscript would present a compelling and well-polished contribution to the field.

Comments on the Quality of English Language

A thorough language proofreading is necessary. The manuscript has moderate issues with English usage. There are recurring grammatical mistakes, improper spacing, missing connecting words, and unclear sentence structures that impact readability.

Author Response

Thank you very much for taking time to carefully review our article and we're sorry for late reply. Thanks again for thoroughly  proofreading of our article. We adopted these corrections in our article to ensure readability. What's more ,as far as we know,  five suggestions concerning methodology are listed in your review. We have carefully considered them and thus we have made some revisions in our article with critical revisions filled orange. In this letter, we will provide detailed response to your comments as follows: 

Response 1: We agree that parameter ablation and analysis on parameter initial dropout rate and initial threshold value can make our model more comprehensive and convincing, so we add subsection 5.2 to describe this point.

Comment: Definition of hyperparameter b_0 in equation 7 is not clearly mentioned. It’s been stated as initial dropout rate at the beginning of when random dropout is activated. Also, some ablations or rationale behind the choice of b_0 value of 0.24 is missing. 

Response : We agree that parameter ablation and analysis on parameter initial dropout rate can make our model more comprehensive and convincing, so we add subsection 5.2 to describe this point. Hyperparameter b_0 exactly means initial dropout rate at the beginning of when random dropout is activated. At the beginning of training, dynamic modality dropout is not activated to avoid our model converging to suboptimal performance. Then, after 20% of training process, dynamic-decreasing visible dropout is activated, and in this period the dropout rate is b_0. Meanwhile, sensitivity analysis of initial threshold value is also applied in subsection 5.2.

Comment: For Threshold Masking (TM) in DMDTM strategy, equations 8 & 9 and related text require better and simplified explanation and flow to understand. It is difficult to follow in the current version. 

Response: We think that defination and explanation of equations 8 could be improved, so we have made some revisions in subsection 3.4 which are filled orange. We think that equations 9 is easy to understand and do not need additional explanation.

Comment: In equation 8, how m is set equal to 0.5. Provide some discussion/explanation about the choice. 

Response: We have made explanation on the choice of hyperparameter m in our revised article, also filled orange. We made some experiments and found that our model is not quite sensitive to hypermeter m, so detailed matrix ablation of both m and initial threshold is not pragmatic.

Comment: Equations 17 and 18, along with their supporting text, feel disconnected from the flow. 

Response: Sorry for misplacement of these equations. Revisions of our article have included them.

Comment:There is no mention of how spatial alignment between RGB and IR channels is ensured or handled.  

Response: We mentioned spatial alignment in "Worse still, the misalignment of two modalities exacerbates the tendency of multispectral method to choose the visible modality instead of combining information of both." in our original article. On second thought, this statement is not supproted by solid evidence or works, so we should not include them in our revised article.

Round 2

Reviewer 1 Report

Comments and Suggestions for Authors

It is evident that the authors have carefully considered and effectively addressed all the concerns that were previously raised.  Therefore, I recommend accepting this manuscript.